# Dimethyl Fumarate-Loaded Transethosomes: A Formulative Study and Preliminary Ex Vivo and In Vivo Evaluation

**DOI:** 10.3390/ijms23158756

**Published:** 2022-08-06

**Authors:** Francesca Ferrara, Mascia Benedusi, Franco Cervellati, Maddalena Sguizzato, Leda Montesi, Agnese Bondi, Markus Drechsler, Walter Pula, Giuseppe Valacchi, Elisabetta Esposito

**Affiliations:** 1Department of Chemical, Pharmaceutical and Agricultural Sciences, University of Ferrara, I-44121 Ferrara, Italy; 2Department of Neurosciences and Rehabilitation, University of Ferrara, I-44121 Ferrara, Italy; 3Department of Life Sciences and Biotechnology, University of Ferrara, I-44121 Ferrara, Italy; 4Bavarian Polymer Institute (BPI) Keylab “Electron and Optical Microscopy”, University of Bayreuth, D-95440 Bayreuth, Germany; 5Animal Science Department, Plants for Human Health Institute, NC Research Campus, NC State University, Kannapolis, NC 28081, USA; 6Department of Environmental Sciences and Prevention, University of Ferrara, I-44121 Ferrara, Italy; 7Department of Food and Nutrition, Kyung Hee University, Seoul 02447, Korea

**Keywords:** dimethyl fumarate, cryogenic transmission electron microscopy, wound healing, transethosomes, transdermal delivery

## Abstract

In this study, transethosomes were investigated as potential delivery systems for dimethyl fumarate. A formulative study was performed investigating the effect of the composition of transethosomes on the morphology and size of vesicles, as well as drug entrapment capacity, using cryogenic transmission electron microscopy, photon correlation spectroscopy, and HPLC. The stability of vesicles was evaluated, both for size increase and capability to control the drug degradation. Drug release kinetics and permeability profiles were evaluated in vitro using Franz cells, associated with different synthetic membranes. The in vitro viability, as well as the capacity to improve wound healing, were evaluated in human keratinocytes. Transmission electron microscopy enabled the evaluation of transethosome uptake and intracellular fate. Based on the obtained results, a transethosome gel was further formulated for the cutaneous application of dimethyl fumarate, the safety of which was evaluated in vivo with a patch test. It was found that the phosphatidylcholine concentration affected vesicle size and lamellarity, influencing the capacity to control dimethyl fumarate’s chemical stability and release kinetics. Indeed, phosphatidylcholine 2.7% *w*/*w* led to multivesicular vesicles with 344 nm mean size, controlling the drug’s chemical stability for at least 90 days. Conversely, phosphatidylcholine 0.9% *w*/*w* resulted in 130 nm sized unilamellar vesicles, which maintained 55% of the drug over 3 months. These latest kinds of transethosomes were able to improve wound healing in vitro and were easily internalised by keratinocytes. The selected transethosome gel, loading 25 mg/mL dimethyl fumarate, was not irritant after cutaneous application under occlusion, suggesting its possible suitability in the treatment of wounds caused by diabetes mellitus or peripheral vascular diseases.

## 1. Introduction

The main function of the skin is to protect our body from external stressors; nevertheless, the skin can be susceptible to acute external injuries and trauma, resulting in a loss of barrier function, exposing the organism to infections, thermal dysregulation, and fluid loss [1,2,3]. In addition, impaired wound healing can be caused by some chronic conditions, such as diabetes mellitus or peripheral vascular diseases [4,5]. Dimethyl fumarate (DMF) is a fumaric acid ester derivate used as a first-line systemic psoriasis treatment, displaying an acceptable safety profile for long-term therapy [6,7]. DMF is also employed in the treatment of relapsing–remitting multiple sclerosis due to being a potent immunomodulator [8]. It is able to restore cutaneous T-cell lymphoma in vitro and in vivo in preclinical models [9], and it can suppress the innate inflammatory responses of glial cells by inhibiting nuclear factor kappa B (NF-κB) activation [10]. Moreover, DMF has demonstrated its suitability to improve wound healing under diabetic conditions [11].

Besides the traditional approaches for wound treatment, such as medical gauze, thin film dressing, foams, spongy materials, and hydrogels, recently, different polymer composite wound dressings have been proposed [12,13,14]. These composite materials are based on natural or synthetic biocompatible polymers suitable for tissue regeneration, filled with nanoparticles conferring antimicrobial activity. Particularly, nanoparticles based on silver, gold, copper, zinc oxide, and titanium dioxide can be employed [13]. Moreover, polymeric nanofibers, encapsulating biological molecules or drugs, have been proposed [12]. The small size and the physical–chemical properties of nanoplatforms enable the promotion of drug penetration into the wounds and/or prolong the drugs’ therapeutic effects. Nonetheless, some drawbacks can be related to wound healing composite materials, such as the cytotoxicity of inorganic nanoparticles, as well as the complex and costly fabrication modalities [15]. In this respect, phospholipid-based colloidal systems appear as an attractive alternative approach. For instance, recently liposomes entrapping bacteriophage cocktails were proposed to treat diabetic wound infections [16], while curcumin-loaded ethosomes were investigated as a veterinary approach for wound management [17]. Furthermore, Mombeiny et al. proposed ethosomes to improve piroxicam delivery in the treatment of wound healing [18]. Ethosomes are colloidal nanosystems based on phosphatidylcholine (PC) organised as multilamellar vesicles embodying high amounts of ethanol (20–45%) [19]. These nanosystems are more thermodynamically stable with respect to liposomes since the presence of ethanol stabilises the vesicles while improving their softness and the capability to load lipophilic drugs [20,21,22]. Despite the ethosomes’ transdermal potential, the authors described the need to combine their topical application to iontophoresis to achieve a transdermal delivery [18]. To further increase ethosomes’ transdermal effects, avoiding the use of physical methods such as iontophoresis, transethosomes (TETs) have been developed [23,24,25]. In TETs, the composition of ethosomes is implemented by the presence of edge activators, such as surfactants, allowing improvement in both drug encapsulation efficiency and vesicles’ penetration potential [24,25,26]. For instance, polysorbate 80 can be added to PC in an ethanol solution, resulting in vesicles with size, morphology, and deformability suitable for transdermal penetration once applied to the skin [27,28]. PC confers to the vesicle biocompatibility and affinity with *stratum corneum* lipids, while ethanol in high concentrations acts as a penetration enhancer, promoting TET passage through the skin. Moreover, ethanol can help prevent wound infections [18].

In this regard, the present work is aimed at investigating the suitability of transethosomal formulation to promote DMF diffusion through the skin, improving wound healing. In terms of a TET clinical application, there is a need to understand vesicle fate and interactions with cells, such as keratinocytes, as well as their in vivo behaviour and toxicity [12]. Therefore, from a technological point of view, physicochemical characterisation of TET is required; indeed, size and morphology can influence vesicle uptake, as well as the modality of drug release and diffusion. Thus, a formulative study was conducted, evaluating the influence of PC concentration on the morphology, size, encapsulation efficiency, and stability of DMF-loaded TET. On the other hand, from a biological perspective, cell viability tests are pivotal to verifying the vesicles’ capability to control the possible toxic effects of encapsulated drugs. In this respect, biological studies were performed to detect the safe DMF concentration for HaCaT human keratinocytes, and TET suitability to promote wound healing. Lastly, a TET hydrogel was specifically designed as a safe biomaterial suitable for wounds. Indeed, due to their consistency and high water content, hydrogels can be comfortably applied to the wound interface, creating a moist environment able to adsorb exudate while acting as barriers against microorganisms [12].

## 2. Results

### 2.1. Preparation of Transethosomes

In order to promote the transdermal delivery of DMF, TETs were designed. These lipid colloidal systems can be obtained by adding water to ethanolic solutions of PC as the main vesicle component (Table 1) [28]. In this study, different PC amounts were employed, selected on the basis of previous investigations performed by our group [25,26], namely 0.9 and 2.7%, *w*/*w*, while the T80 concentration was 0.3% *w*/*w*, which is suitable to obtain stable vesicles (Table 1). TET_0.9_ and TET_2.7_ appeared as homogeneous dispersions, more translucent in the case of the lower PC concentration. DMF (500 mg/mL) was loaded in TETs via previous solubilisation in PC/T80 ethanol solutions, resulting in whitish homogeneous dispersions free from separation phenomena.

### 2.2. Morphology of Transethosomes

The morphology of TET was investigated using cryo-TEM. As shown in Figure 1, spherical vesicles were obtained, unilamellar in the case of TET_0.9_-DMF (Figure 1a), comprising PC 0.9% *w*/*w*.

In the case of TET_2.7_-DMF, the three-fold PC concentration resulted in a differentially organised double layer, leading to heterogeneous vesicles, unilamellar as well as multilamellar, and multivesicular vesicles. In the case of TET_0.9_-DMF_0.5_, the intercalation of the T80 oleate chain within the lipid bilayer could hamper the formation of multilamellar vesicles, while in the case of TET_2.7_-DMF_0.5_, a three-fold increase in PC concentration, under the same amount of T80, improved the lamellar organisation, resulting in multilamellar and multivesicular vesicles.

### 2.3. Size Distribution of Transethosomes

The size distribution parameters of TETs measured with PCS are reported in Table 2. The mean diameters were strongly influenced by PC content; indeed, in the case of PC 0.9% *w*/*w*, the size of vesicles ranged between 130 and 144 nm, while in the case of PC 2.7%, the Z average more than doubled, ranging between 344 and 350 nm. DMF presence slightly reduced the mean diameters of the vesicles. The larger size of TET_2.7_-DMF_0.5_ vesicles is due to the PC’s self-organisation in multilamellar and oligolamellar vesicles compared with the TET_0.9_-DMF_0.5_ unilamellar structure, as observed via cryo-TEM (Figure 1). The slight increase in vesicle size due to DMF presence could be related to its possible placement within the TET vesicles, near the head groups of PC at the interface between the glycerol groups and the apolar lipid chains. This positioning can increase the average distance among the PC molecules constituting the bilayer of the vesicles [29].

### 2.4. DMF Entrapment Capacity (EC)

To evaluate the EC of DMF in the vesicular formulations, the lipid phase was separated from the aqueous one via ultrafiltration, before dissolution in ethanol to promote vesicle disaggregation. DMF quantification using HPLC confirmed the total recovery of the drug in the lipid phase; the EC values were indeed 99 ± 1% and 98.5 ± 1.5%, respectively, for TET_0.9_-DMF and TET_2.7_-DMF.

### 2.5. Stability Evaluation

TET samples were stored in the light at 22 °C for 3 months to evaluate the size and chemical stability. Specifically, vesicles’ mean diameters were monitored via PCS to gain information on size stability. As reported in Figure 2a, the Z average values increased as a function of PC concentration and DMF presence. In addition, the size increase ratio (SIR) was very higher in TET_0.9_ (SIR 80.6%) than in TET _2.7_ (SIR 7.8%). In the case of TET_0.9_-DMF, the drug presence allowed better control of the size increase (SIR 68.0%) relative to the empty TET. Conversely, in the case of TET_2.7_-DMF, the increase in size was more pronounced (SIR 21.9%) than the corresponding TET produced in the absence of DMF. Dispersity indexes did not increase and were always below 0.2, indicating a homogeneous size distribution.

To evaluate the capability of TET_0.9_-DMF and TET_2.7_-DMF to control DMF degradation, the residual drug in the vesicles was analysed and compared with a DMF solution of 0.5 mg/mL in ethanol/water (30:70) (SOL-DMF). In general, vesicular systems were able to control DMF degradation compared with SOL-DMF. Indeed, in this latest case, the complete degradation of DMF occurred in 85–90 days, while after 90 days, in the cases of TET_0.9_-DMF and TET_2.7_-DMF, the residual DMF concentrations were, respectively, 55% and 92% relative to the initial content.

### 2.6. In Vitro Release Test (IVRT)

Franz cells associated with synthetic membranes composed of PTFE were employed to compare the DMF release kinetics from vesicular systems with those of SOL-DMF. The PTFE porous synthetic membrane, mounted between the upper and lower compartments of the Franz cell, simply represents a physical support structure, avoiding the mixing of donor and receptor phases. As shown in Figure 3, both TET_0.9_-DMF and TET_2.7_-DMF enabled the control of DMF release compared with SOL-DMF.

DMF release rates R, calculated considering the slope of the linear part of the profiles (Appendix A), are reported in Table 3. The R_DMF_ values, as well as A_DMF_, followed the order SOL-DMF > TET_0.9_-DMF > T-ET_2.7_-DMF. It is conceivable that, in the case of TET_2.7_-DMF, the higher PC concentration could more strictly retain the DMF, restraining its release compared with TET_0.9_-DMF. Accordingly, T_lag_ values followed the opposite order, i.e., TET_2.7_-DMF > TET_0.9_-DMF > SOL-DMF.

### 2.7. In Vitro Permeation Test (IVPT)

Franz cells associated with the synthetic membrane Strat-M^®^ were employed to study the effect of the type of TETs on DMF permeation. Strat-M^®^ is a multilayered polymeric membrane system suitable to mimic the skin. Indeed, it constitutes two polyether sulfone layers overlapped to one polyolefin bottom layer, resulting in a skin-like, tortuous, porous structure [30]. Notably, this membrane system is impregnated with synthetic lipids to further provide a skin resemblance, characterised by hydrophilic and lipophilic compartments, which confer barrier properties. The passive kinetic process of drug permeation from the vehicle into the skin or membrane system comprises two main phases [31,32,33]. The first phase, referring to the drug release from the vehicle, can be described by the partition coefficient (P), reflecting the preferred drug distribution in the skin/membrane or the vehicle. The second phase refers to the diffusion of the drug through the lipophilic and hydrophilic layers of the skin/membrane, depending on the physicochemical drug characteristics and the composition of the vehicle. This phase can be typified by the diffusion (D) and permeability (Kp) coefficients as permeation parameters. Figure 4 demonstrates the DMF permeation profiles from TET_0.9_-DMF, T-ET_2.7_-DMF, and SOL-QT through Strat-M^®^, while the permeation parameters are reported in Table 4.

As depicted in Figure 4, the fastest kinetic was found in the case of TET_0.9_-DMF, followed by SOL-DMF and T-ET_2.7_-DMF. TET_0.9_-DMF displayed the highest Kp, P, and A_DMF_ values, as reported in Table 4. Notably, in the case of TET_0.9_-DMF, Kp was 1.65-fold higher than T-ET_2.7_-DMF, while the *p* value was double.

### 2.8. Cell Viability

To determine the concentration of DMF suitable for a cutaneous application, an MTT viability test was conducted on HaCaT cells treated with DMF solution (SOL-DMF).

HaCaT cells treated with SOL-DMF at concentrations lower than 2.5 µg/mL for 24 h did not display any changes in cell viability; DMF at a concentration of 5–10 µg/mL just led to a slight increase in cell mortality, whereas a significant effect on cell viability was displayed between 20 and 80 µg/mL of DMF (Figure 5a). Based on the cytotoxicity results, and to determine the effects of TETs on keratinocyte viability, the MTT test was performed on HaCaT cells treated with the different TET formulations containing DMF at non-cytotoxic concentrations (0.5–10 µg/mL). As shown in Figure 5b, for both TET_0.9_-DMF and TET_2.7_-DMF, 2.5 mg/mL was the highest dose not affecting cell viability. Therefore, 2.5 μg/mL was selected for the subsequent assays.

### 2.9. Scratch Assay

To test the capability of TET_0.9_ and TET_2.7_ loaded with DMF to promote wound healing, a scratch assay was performed on HaCaT cells treated or untreated with TET_0.9_-DMF and TET_2.7_-DMF (Figure 6). After 24 h, TET_0.9_-DMF was more effective in facilitating wound closure than the control condition, the unloaded formulation, and TET_2.7_-DMF.

### 2.10. Transmission Electron Microscopy

The uptake of TET in keratinocytes incubated with TET_0.9_-DMF and TET_2.7_-DMF was investigated with TEM analysis at different time points (2, 6, and 24 h). As shown in Figure 7a–c, TET_0.9_-DMF appeared as roundish electron-dense vesicles, showing a dark rim, corresponding to the PC double layer, and a grey core, indicated by arrows. Remarkably, after 2 h incubation (Figure 7a), many vesicles were detected in the intracellular milieu. The presence of TET_0.9_-DMF was still detectable after 6 h, as indicated by the black arrow in Figure 7b, showing some vesicles in the proximity of mitochondria, as previously found [34]. After 24 h, very few vesicles were recognizable (Figure 7c). Conversely, in the case of TET_2.7_-DMF, the absence of electron-dense vesicles inside the cells indicated the inability of keratinocytes to uptake these kinds of vesicles (Figure 7d–f). Indeed, after 2 h incubation (Figure 7d), no vesicles were present inside the cell, while after 6 h, some roundish structures were found in the cell (Figure 7e). Lastly, after 24 h, few large structures were detectable on the keratinocyte surface (Figure 7f), possibly deriving from the fusion of TET_2.7_-DMF that were not able to penetrate.

### 2.11. TET_0.9_-DMF50 Formulation

Based on biological studies, TET_0.9_-DMF was more efficient TET_2.7_-DMF to induce wound healing in keratinocytes. In addition, TEM analyses confirmed the uptake of TET_0.9_-DMF in keratinocytes rather than TET_2.7_-DMF. Thus, TET_0.9_-DMF was selected for skin application. Since the highest safe concentration that emerged from cell viability studies was 2.5 mg/mL, relatively lower than the amount of diffused DMF via IVPT (A_DMF_), DMF loading in TETs was 10-fold decreased, obtaining TET_0.9_-DMF50 (DMF 50 mg/mL). The TET_0.9_-DMF50 morphology, size distribution parameters, and EC values were comparable to the parameters of TET_0.9_-DMF loaded with DMF 500 mg/mL, as reported in Appendix A and Appendix A. The DMF permeability kinetics, determined by Franz cells associated with Strat-M^®^, are reported in Figure 8. Apart from the lower A_DMF_ (18 mg/mL), Kp, P, and D were almost identical to the values obtained using the more DMF-concentrated TET_0.9_-DMF (Appendix A).

### 2.12. TET Gel Formulation

To obtain a semisolid vehicle suitable for the cutaneous application, TET_0.9_-DMF50 were thickened via dilution with an x-gum gel, leading to the final concentration of 25 mg/mL (TET_0.9_-DMF25-gel) and 10 mg/mL (TET_0.9_-DMF10-gel) DMF. Both gels were characterised by their semisolid consistency suitable for administration on the skin and homogeneous features. The comparison between the DMF diffusion kinetics of TET_0.9_-DMF25-gel and TET_0.9_-DMF10-gel is reported in Figure 8. As expected, the rates of DMF diffusion were slower than that of TET_0.9_-DMF50 and were affected by DMF concentration. Kp values were 26 and 17 (cm h^−1^ 10^−3^) for TET_0.9_-DMF25-gel and TET_0.9_-DMF10-gel, respectively. In the case of TET_0.9_-DMF25-gel, A_DMF_ was 7.5 mg/mL, while the amount of DMF diffused after 24 h was 2.2 mg/mL in the case of TET_0.9_-DMF10-gel (Appendix A). This DMF concentration was below the cytotoxic concentration for keratinocytes, as reported above. For this reason, TET_0.9_-DMF10-gel was selected for further in vivo studies.

### 2.13. Patch Test

In order to verify the safeness of the TET gel once applied to the skin, a patch test was performed. Both the empty TET_0.9_-gel and TET_0.9_-DMF10-gel were evaluated. The formulations applied for 48 h under occlusive conditions on the healthy skin of 20 volunteers resulted in a 0.15 average irritation index, suggesting a classification of the gels as “non-irritating”.

## 3. Discussion

The present study highlighted the potential of TETs for the transdermal administration of DMF. TET preparation, conducted via a low-energy-consuming, cold method, enabled us to obtain dispersions using a simple colloidal association of PC/T80 in ethanol/water, resulting in the formation of vesicles. It is well known that surfactants can spontaneously form different crystalline structures. Specifically, the supramolecular assembly of amphiphiles such as PC, characterised by two hydrophilic tails and one hydrophobic head group, results in bilayered vesicles. The shape and size of vesicles are mainly influenced by the molecular packing parameter of the surfactant, a concept based on the geometrical ratio between its tail volume and the product of the head area and the tail length [35]. T80 was previously selected by our group for transethosome production, due to its capability to control both the size distribution and stability of vesicles [28,34]. Remarkably, PC concentration strongly affected the morphology and size distribution of the vesicles. Indeed, it has been observed that unilamellar vesicles often occur in diluted surfactant solutions, while multivesicular and oligovesicular vesicles are usually found in more concentrated ones [36]. Concerning the influence of PC concentration on vesicle size, different authors described the enlargement of mean diameters due to an increase in PC concentration [29,37]. Specifically, the TET based on PC 2.7% *w*/*w* led to multivesicular vesicles, with the Z average value more than doubled compared with that of the unilamellar TET based on PC 0.9% *w*/*w*. These differences strongly affected the performance of the different TETs. It is likely that the more structured vesicles produced by the higher PC amount could better accommodate DMF within the multilayers. Indeed, although both vesicular nanosystems improved DMF stability compared with SOL-DMF, TET_2.7_-DMF enabled a longer control of DMF degradation as well as release kinetics than TET_0.9_-DMF. On the other hand, TET_0.9_-DMF improved DMF in vitro permeability compared with SOL-DMF and TET_2.7_-DMF, suggesting the capability of unilamellar vesicles to enhance DMF diffusion through the membrane, resulting in the highest DMF amount diffused after 24 h. Conversely, the lowest *p* value calculated in the case of TET_2.7_-DMF indicated a preferential DMF distribution towards the multivesicular vesicles, rather than the membrane, as recently found in the case of TETs designed for quercetin’s transdermal delivery [29].

The cell viability test enabled the selection of the safe DMF concentration, revealing that neither the 2.5 µg/mL DMF solution nor the 2.5 µg/mL TET_0.9_-DMF vesicles affected cellular viability after 24 h of treatment.

Moreover, the biological data agreed with technological findings. It has been shown that DMF can exert a dual role, proliferative or antiproliferative, depending on its concentration. For instance, at a low dose (1 mM), DMF induces cell growth, while at a high dose (100 mM), it significantly inhibits cell proliferation [38]. The wound healing assay confirmed the proliferative effect of DMF at a low dose (2.5 µg/mL) delivered by TET_0.9_-DMF, supporting the hypothesis of a higher penetration enhancement effect exerted by unilamellar vesicles. This result was corroborated by TEM observations that provided evidence of the uptake of TET_0.9_-DMF by HaCaT cells already after 2 h of incubation. The capability of TETs to pass through the cell membrane is ascribable to the presence of ethanol that exerts a penetration enhancement effect and improves vesicle malleability. Nonetheless, this study demonstrated that the supramolecular organisation of PC, as well as the size of TET vesicles, can affect this uptake. Indeed, the large diameter of T-ETO_2.7_-DMF multivesicular vesicles hampered their internalisation in HaCaT cells. This evidence agrees with previous findings demonstrating that the particle size of lipidic vesicles strongly affects the bioactive delivery to the skin [39]. The presence of intact TET_0.9_-DMF vesicles after 2 h, followed by their progressive disappearance over time, suggests a gradual vesicle degradation, occurring through physiological pathways [34], and a consequent DMF release, as evidenced by the wound healing effect.

## 4. Materials and Methods

### 4.1. Materials

Dimethyl fumarate (dimethyl (E)-but-2-enedioate, DMF), polysorbate 80 (polyoxyethylene sorbitan monooleate, tween 80, T80), and xanthan gum (x-gum) were purchased from Merck Life Science S.r.l. (Milan, Italy). The soybean lecithin (PC) (90% phosphatidylcholine) was Epikuron 200 from Lucas Meyer (Hamburg, Germany). Polytetrafluoroethylene (PTFE, Whatman^®^) (pore size 200 nm, thickness 155.6 μm), and Strat-M^®^ (thickness 316.9 μm) [40] membranes were purchased from Merck Life Science S.r.l. (Milan, Italy). Solvents were of HPLC grade, and all other chemicals were of analytical grade.

### 4.2. Transethosome Preparation

TET preparation was obtained via the cold method [29,34]. Briefly, PC was solubilised in ethanol (30 or 90 mg/mL) under stirring at 750 rpm (IKA RCT basic, IKA^®^-Werke GmbH & Co. KG, Staufen, Germany); afterwards, the surfactant T80 was added to the PC solution up to a final 1 % *w*/*v* concentration. After complete solubilisation, bidistilled water was added dropwise to the PC/T80 solution up to a final 70:30 (*v*/*v*) water/ethanol ratio. The magnetic stirring was maintained for 30 min. To load DMF in TETs, the drug (0.5 mg/mL) was solubilised in the PC/T80 ethanol solution before adding water. To obtain TET_0.9_-DMF50, 0.05 mg/mL of DMF was employed.

### 4.3. Photon Correlation Spectroscopy (PCS)

Vesicle size distribution was measured using a Zetasizer Nano-S90 (Malvern Instr., Malvern, UK) with a 5 mW helium-neon laser and a wavelength output of 633 nm. Measurements were performed at 25 °C at a 90° angle and a run time of at least 180 s. Samples were diluted with bidistilled water in a 1:20 *v*/*v* ratio. Data were analysed using the “CONTIN” method [41]. Measurements were performed thrice for 3 months after TET production, on TETs stored at 22 °C, calculating the mean ± standard deviation (s.d.). The size increase ratios (SIRs) of TETs were expressed by calculating the difference between the Z average mean diameter of TETs stored for 3 months and the Z average mean diameter of TETs measured the day after preparation, as follows:(1)$$\mathrm{SIR}=\frac{{Z\;\mathrm{Average}}_{\mathrm{day}\;90}-{Z\;\mathrm{Average}}_{\mathrm{day}\;1}}{{Z\;\mathrm{Average}}_{\mathrm{day}\;1}}\times 100$$

The statistical differences were evaluated by Student’s *t*-test and GraphPad Prism 9 software (GraphPad Software Inc. San Diego, CA, USA), considering values of *p* < 0.05 as statistically significant.

### 4.4. Cryo-Transmission Electron Microscopy (Cryo-TEM)

For cryo-TEM analyses, samples were vitrified following a method previously reported [42]. Specifically, a 2 μL aliquot of the sample was put for a few seconds on a lacey carbon filmed copper grid (Science Services, München, Germany). After removing most of the liquid with a blotting paper, a thin film stretched over the lace holes was obtained. Vitrification was achieved by the rapid immersion of specimens into liquid ethane cooled to approximately 90 K (−180 °C) by liquid nitrogen in a temperature-controlled freezing unit (Leica EMGP, Leica, Germany). The sample preparation procedure was conducted at a controlled, constant temperature in a Leica EMGP chamber. The vitrified specimens were transferred to a Zeiss/Leo EM922 Omega EFTEM (Zeiss Microscopy GmbH, Jena, Germany) transmission electron microscope using a cryo-holder (CT3500, Gatan, Munich, Germany). During microscopy observations, the samples’ temperature was kept below 100 K. Specimens were examined with reduced doses of ≈ 1000–2000 e/nm^2^ at 200 kV. Zero-loss, filtered images (ΔE = 0 eV) were recorded using a CCD digital camera (Ultrascan 1000, Gatan, Munich, Germany) and analysed with the GMS 1.9 software (Gatan, Munich, Germany).

### 4.5. Evaluation of DMF Entrapment Capacity in Transethosome

The entrapment capacity (EC) of DMF in TETs was determined via ultrafiltration, 24 h after preparation using a centrifugal filter device (Microcon centrifugal filter unit YM-10 membrane, NMWCO 10 kDa, Sigma-Aldrich, St. Louis, MO, USA) and HPLC analysis, as reported below. Specifically, 500 μL of DMF-loaded TET were poured into the sample’s reservoir part of the device and subjected to ultrafiltration (Spectrafuge™ 24D Digital Microcentrifuge, Woodbridge, NJ, USA) at 4000 rpm for 15 min. Afterwards, both retentate and filtrate fractions were withdrawn, respectively, from the sample’s reservoir part and the vial, and diluted with ethanol (1:10, *v*/*v*). Before the HPLC analysis, the diluted retentate was stirred for 30 min and filtered with nylon syringe membranes (0.22 μm pore diameter), while the filtrate fraction was analysed as such. The entrapment capacity (EC) was determined as follows:EC = DMF /T_DMF_ × 100(2)
where DMF is the amount of the drug measured with HPLC, and T_DMF_ is the total amount of DMF employed for T-ETO preparation. In addition, for chemical stability studies, the DMF content was evaluated monthly for 3 months on the whole TET dispersion stored at 22 °C.

### 4.6. Franz Cell Diffusion Experiments

Franz cells (orifice diameter 0.9 cm; PermeGear Inc. Hellertown, PA, USA) were employed for IVRT and IVPT. Notably, PTFE membranes were used for IVRT, while Strat-M^®^ membranes were employed for IVPT [30,31]. Both for IVRT and IVPT, the samples of dried membranes were rehydrated via immersion in ethanol/water 50:50, *v*/*v* for 1 h, before assembling in Franz-type diffusion cells. The receptor compartment of the cell contained 5 mL of ethanol:water 50:50, *v*/*v* in order to assure sink conditions, stirred at 500 rpm by a magnetic bar, and thermostated at 32 ± 1 °C during the experiments [29]. A total of 500 μL of DMF-loaded T-ETO or DMF ethanolic solution (ethanol:water 30:70, *v*/*v*) (DMF 0.5 mg/mL) (SOL-DMF) was placed on the membrane surface in the donor compartment which was sealed afterwards to avoid evaporation. At predetermined time intervals between 1 and 24 h, samples (0.5 mL) of the receptor phase were withdrawn and analysed with HPLC to evaluate the DMF content. Each removed sample was replaced with an equal volume of simple receptor phase. The DMF concentrations were determined 6 times in independent experiments, and the mean values ± s.d. were calculated. By the end of the Franz cell experiment, the membranes were cut, put in vials containing 1 mL of ethanol, and subjected to ultrasonication (Branson, Bransonic^®^ M Mechanical Bath 3800, Emerson, St. Louis, MO 63136, USA) for 15 min. Afterwards, the suspension was filtered with nylon syringe membranes and analysed using HPLC. The amount of DMF associated with the membranes (M_DMF_) was calculated.

#### 4.6.1. In Vitro Release Test (IVRT)

For data analysis, in the case of IVRT, the DMF amount (μg/cm^2^) was plotted as a function of the square root of time. To compare the DMF release kinetics from the different dispersions, the following parameters were evaluated: “R_DMF_”, the slope of the cumulative amount of DMF released versus the square root of time; lag-time “T_lag_”, extrapolated from the intercept of the release profile with x-axis; and “A_DMF_”, the cumulative amount of DMF released at the last sampling time (8 h).

#### 4.6.2. In Vitro Skin Permeation Test (IVPT)

In the case of IVPT, to analyse the data, Fick’s law was considered since it describes the steady-state permeation through the skin, assuming that, under sink conditions, the drug concentration in the receptor compartment is negligible compared with that in the donor compartment [43]. The steady-state flux of drug per unit area “Jss” is described as
Jss = P × Cd × D/e(3)
where P is the partition coefficient, Cd is the drug concentration in the donor compartment, D is the DMF diffusion coefficient, and e is the thickness given by the supplier or measured. Accordingly, DMF permeability coefficients, “Kp” and T_lag_ values, were calculated considering the steady-state portion of DMF cumulative penetration profiles versus time. The slope of the linear part of the curves yielded the pseudo-steady-state flux “Jss” (µg/cm^2^/h) [43]. Kp was calculated according to Equation (4):Kp = Jss/Cd(4)

The D value was calculated from Tlag according to Equation (5):T_lag_ = e^2^/6 × D(5)

Lastly, P was calculated considering Equation (6):Kp = D × P/e(6)

### 4.7. HPLC Analysis

HPLC analyses were performed using PerkinElmer, Series 200 HPLC Systems equipped with a micro-pump, an autosampler, and a UV detector operating at 255 nm. A stainless-steel C-18 reverse-phase column (15 × 0.46 cm) packed with 5 μm particles (Hypersil BDS C18 Thermo Fisher Scientific S.p.A., Milan, Italy) was eluted at a flow rate of 1 mL/min, with a mobile phase containing acetonitrile/water 40:60 *v*/*v*.

### 4.8. Biological Activity Studies

#### 4.8.1. Cytotoxicity Study

HaCaT cells were cultured in Dulbecco’s modified Eagle’s medium (DMEM) High Glucose (Lonza, Milan, Italy), supplemented with 10% foetal bovine serum (FBS), 100 U/mL penicillin, 100 μg/mL streptomycin, and 2 mM L-glutamine. Cells were incubated at 37 °C for 24 h in 95% air/5% CO_2_ until 80% confluence.

For cell treatments, the different vehicles were initially dissolved in a cell culture medium to obtain the stock solutions containing DMF 500 µg/mL and then further diluted to reach final concentrations of DMF 0.5–5 μg/mL.

First, the cells were grown in 96-well plates at a density of 2 × 10^4^ cells/well in 200 μL of media for the MTT assay. Seeded cells were exposed to unloaded and DMF-loaded formulations at various DMF concentrations, ranging from 0.5 μg/mL to 5 μg/mL for 24 h. After the complete removal of the treatment to avoid any colour interference, 50 μL of serum-free media and 50 μL MTT (0.5 mg/mL) were added and incubated for 3 h. The insoluble purple formazan crystals were then dissolved in 100 μL of DMSO at 37 °C for 15 min. After shaking, the solution absorbance was measured with a spectrophotometer at 590 nm, using 670 nm as the reference wavelength, and, thus, converted into a percentage of viability [44].

#### 4.8.2. Scratch Assay

HaCaT cells were grown to a confluent monolayer on 24-well plates and then were mechanically scratched with a 200 μL sterile pipette tip. Cellular debris were removed by washing off with PBS and, based on the cytotoxicity data of the cells, were immediately treated with 2.5 μg/mL of DMF-loaded and unloaded formulations, followed by incubation at 37 °C. Images of the scratches for each sample were recorded immediately after and 24 h post-scratch. The wound healing rate was analysed with ImageJ software (National Institutes of Health, Bethesda, MD, USA) and compared with the wound area at T_0_ [45,46].

### 4.9. Transmission Electron Microscopy (TEM) Analyses

Keratinocytes treated with TET for 2, 6, and 24 h and untreated cells, used as control, were grown to confluent monolayers on 10 cm Petri dishes. The cells were fixed in 2.5% glutaraldehyde in cacodylate buffer for 3 h at 4 °C and then post-fixed in 1% osmium tetroxide for 2 h at 4 °C, dehydrated in a graded series of alcohol, embedded in Araldite resins, and polymerised in the oven for 48 h at 60 °C. Ultrathin sections of 60 nm were cut with an ultramicrotome (Ultratome Reichert SuperNova Leica, Wien, Austria), stained with uranyl acetate and lead citrate, and examined in a Philips CM100 transmission electron microscope [44].

### 4.10. Preparation of TET Gel

To prepare a viscous gel, x-gum (0.5 %, *w*/*w*) was added to bidistilled water under magnetic stirring for 30 min, up to complete dispersion. TET_0.9_-DMF50 were diluted 1:1 or 1:5 *w*/*w* with the x-gum gel and handily mixed for 10 min. The resulting x-gum concentrations in gels were 0.25 % and 0.4 % *w*/*w*, while DMF concentrations were 25 and 10 mg/mL, respectively.

### 4.11. Patch Test

To evaluate the effect of TET gel applied on human skin, an in vivo irritation test was performed. The occlusive patch test was conducted at the Cosmetology Centre of the University of Ferrara following the basic criteria of the protocols for the skin compatibility testing of potentially cutaneous irritant cosmetic ingredients on human volunteers (SCCNFP/0245/99) [47,48,49,50]. The protocol was approved by the Ethics Committee of the University of Ferrara, Italy (study number: 170583). The test was run on 20 healthy volunteers of both sexes, who gave written consent to the experimentation, excluding subjects affected by dermatitis, with a history of allergic skin reaction or under anti-inflammatory drug therapy (either steroidal or non-steroidal). Ten milligrams of TET gel (TET_0.9_-gel) or TET-gel loaded with DMF (TET_0.9_-DMF10-gel) were placed into aluminium Finn chambers (Bracco, Milan, Italy) and applied onto the skin of the forearm or the back protected by a self-sticking tape. Specifically, samples were directly applied into the Finn chamber with an insulin syringe and left in contact with the skin surface for 48 h. Skin irritative reactions (erythematous and/or oedematous) were evaluated 15 min and 24 h after removing the patch and cleaning the skin from sample residual. Erythematous reactions were sorted into three groups according to the reaction degree: light, clearly visible, and moderate/serious erythema. The average irritation index was calculated as the sum of erythema and oedema scores and expressed according to a scale considering 0.5 as the threshold above which the product is to be classified as slightly irritating, 2.5–5 as moderately irritating, and 5–8 as highly irritating.

## 5. Conclusions

This study demonstrated that TET_0.9_-DMF vesicles are characterised by suitable physicochemical properties for efficient DMF loading and transdermal penetration. Remarkably, in vitro studies indicated that TET_0.9_-DMF vesicles are able to enter HaCaT cells and release DMF, allowing the improvement of wound closure. TET_0.9_-DMF10-gel specifically formulated based on cell viability studies can be safely applied on the skin, as demonstrated by the patch test. These findings pave the way to further perform ex vivo and in vivo studies aimed at verifying the suitability of TET_0.9_-DMF in the treatment of chronic conditions such as diabetes mellitus or peripheral vascular disease.

## Figures and Tables

**Figure 1 ijms-23-08756-f001:**
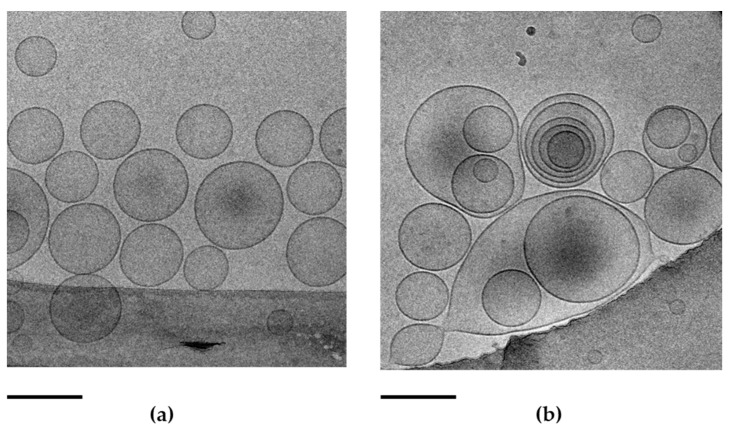
Cryo-TEM images of TET_0.9_-DMF_0.5_ (**a**) and TET_2.7_-DMF_0.5_ (**b**). The bar corresponds to 200 nm in both panels.

**Figure 2 ijms-23-08756-f002:**
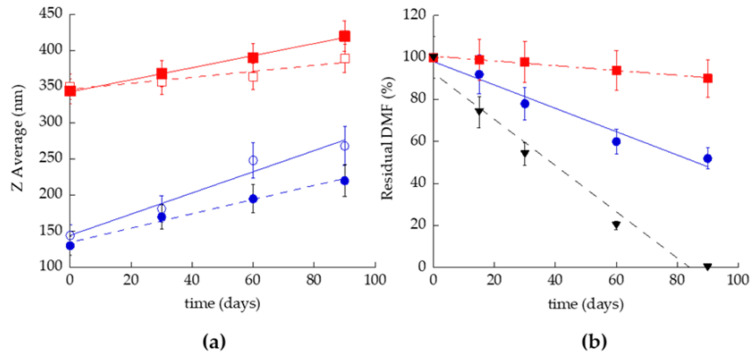
Effect of storage on ethosomes and transethosomes kept at 22 °C for 3 months: (**a**) Z average mean diameters measured via PCS on TET_0.9_ (o), TET_0.9_-DMF (•), TET_2.7_ (□), and TET_2.7_-DMF (▪), *p* values < 0.05; (**b**) DMF residual content in T-ETO_0.9_-DMF (•), T-ETO_0.9_-DMF (▪) and SOL-DMF (▼). The percentage refers to the total DMF content evaluated after sample preparation; *p* values < 0.05.

**Figure 3 ijms-23-08756-f003:**
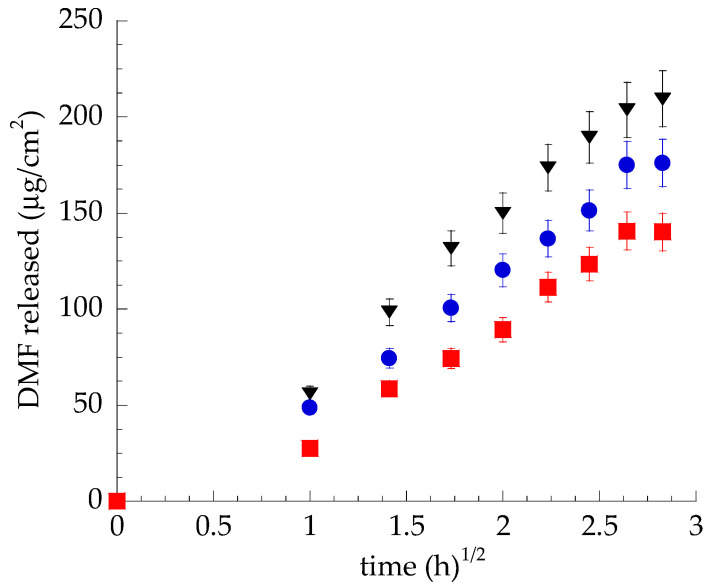
DMF release kinetics from TET_0.9_-DMF (•), TET_2.7_-DMF (▪), and SOL-DMF (▼), as determined by Franz cells associated with PTFE. Data are the mean of 6 independent experiments ± s.d.

**Figure 4 ijms-23-08756-f004:**
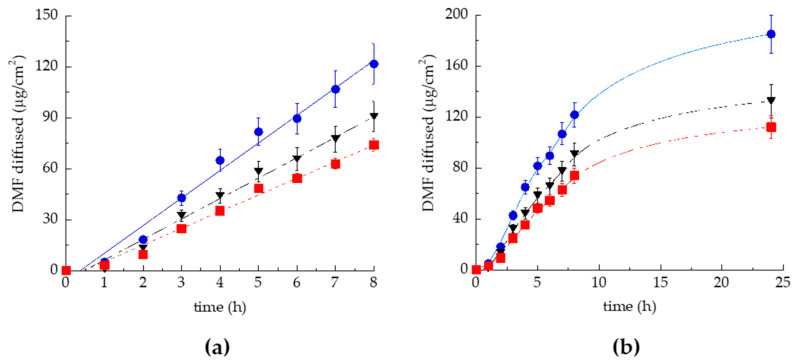
DMF permeability kinetics from TET_0.9_-DMF (•), TET_2.7_-DMF (▪), and SOL-DMF (▼), as determined by Franz cells associated with Strat-M^®^. (**a**) linear part of the kinetic profile (0–8 h); (**b**) 0–24 h kinetic. Data are the mean of 6 independent experiments ± s.d.

**Figure 5 ijms-23-08756-f005:**
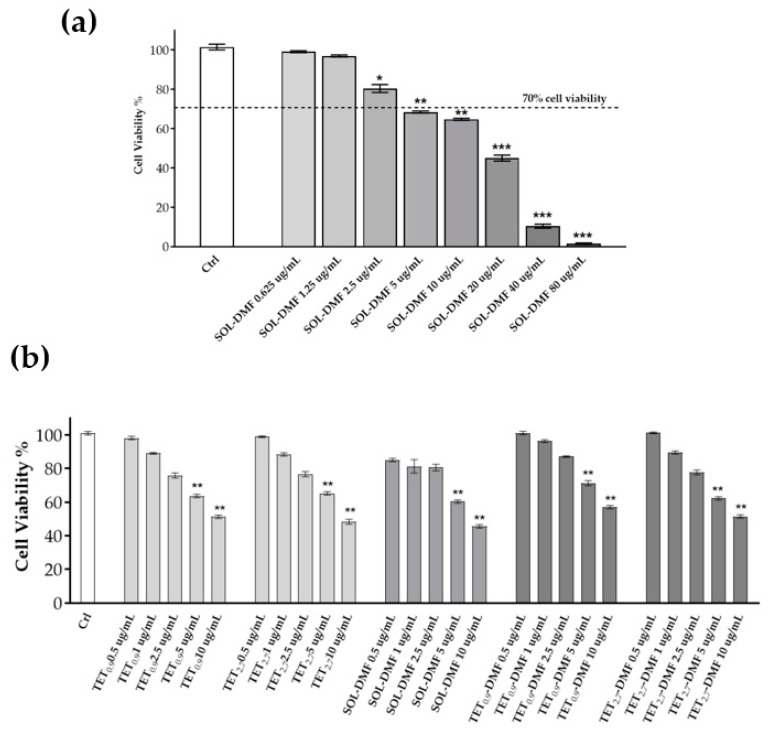
HaCaT cell viability evaluated using an MTT test after 24 h of treatment with SOL-DMF differently diluted (from 0.625 µg/mL to 80 µg/mL) (**a**) and with differently diluted DMF-loaded or unloaded TET_0.9_ and TET_2.7_ (**b**). Data are the results of three independent experiments performed in triplicate. * *p* ≤ 0.01; ** *p* ≤ 0.005; *** *p* ≤ 0.0001 by two-way ANOVA followed by Tukey’s multiple comparison test; treatment vs. ctrl.

**Figure 6 ijms-23-08756-f006:**
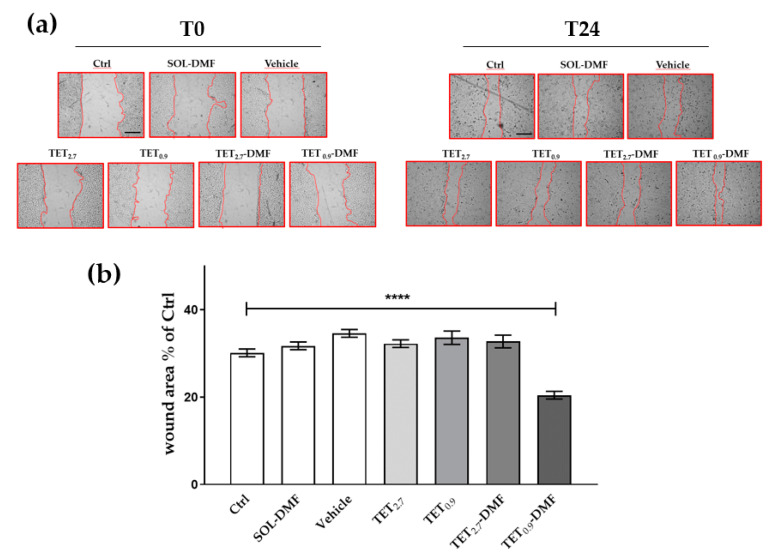
Effect of DMF-loaded or unloaded formulations on the wound closure in HaCaT cells: (**a**) a scratch test was performed on a confluent monolayer of HaCaT cells; different images were taken to measure the wound area right after the scratch (left panel T_0_) and 24 h after (right panel T24) (scale bar 200 μm); (**b**) quantification of the wound area (within the red edges) 24 h after the scratch analysed by using ImageJ software, compared with T_0_. Data are the results of three independent experiments performed in triplicate. **** *p* ≤ 0.0001 using one-way ANOVA followed by Tukey’s multiple comparison test.

**Figure 7 ijms-23-08756-f007:**
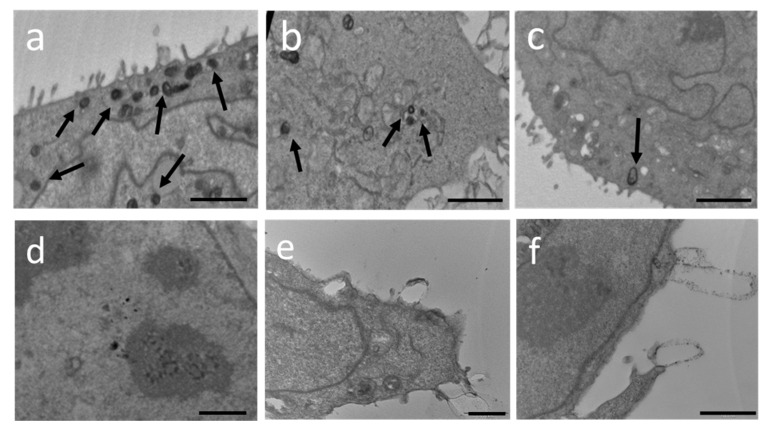
Transmission electron micrographs of keratinocytes incubated with TET_0.9_-DMF (**a**–**c**) or TET_2.7_-DMF (**d**–**f**) for 2 h (**a**,**d**), 6 h (**b**,**e**), and 24 h (**c**,**f**). Bars: 1000 nm (**a**), 2000 nm (**b**,**c**), 3000 nm (**d**), and 1000 nm (**e**,**f**).

**Figure 8 ijms-23-08756-f008:**
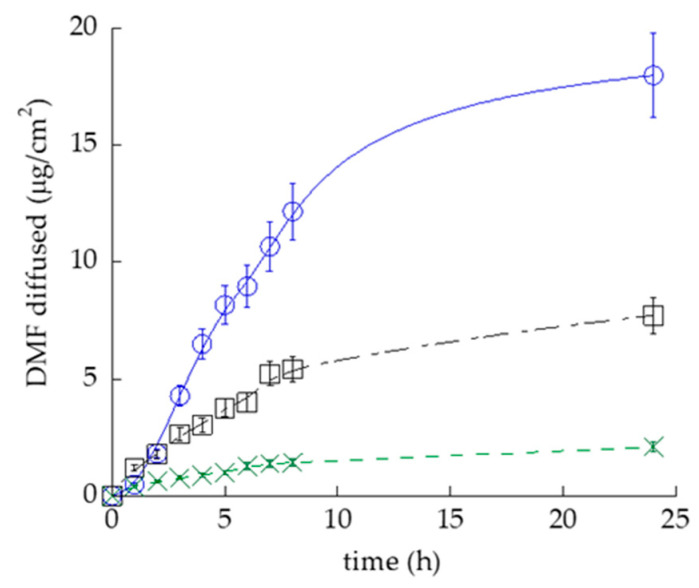
DMF permeability kinetics from TET_0.9_-DMF50 (o), TET_0.9_-DMF25-gel (□), and TET_0.9_-DMF10-gel (x), as determined by Franz cells associated with Strat-M^®^. Data are the mean of 6 independent experiments ± s.d.

**Table 1 ijms-23-08756-t001:** Composition of transethosomes.

Formulation	PC ^1^ % *w*/*w*	Ethanol % *w*/*w*	TW_80_ ^2^ % *w*/*w*	DMF ^3^ % *w*/*w*	Water % *w*/*w*
TET_0.9_	0.89	28.81	0.3	-	70
TET_2.7_	2.69	27.01	0.3	-	70
TET_0.9_-DMF	0.89	28.76	0.3	0.05	70
TET_2.7_-DMF	2.69	26.96	0.3	0.05	70

^1^: Soy phosphatidylcholine; ^2^: polysorbate 80; ^3^: dimethyl fumarate.

**Table 2 ijms-23-08756-t002:** Dimensional distribution parameters of transethosomes, as determined by PCS.

Formulation	Z-Average (nm)	Typical Intensity Distribution (nm)	Dispersity Index
TET_0.9_	144.20 ± 13.13	146.2 (100 %)	0.14 ± 0.02
TET_2.7_	350.40 ± 23.60	724.1 (84%) 166.6 (16 %)	0.23 ± 0.05
TET_0.9_-DMF	130.25 ± 4.50	157.2 (99.6%) 36.15 (0.7%)	0.13 ± 0.01
TET_2.7_-DMF	344.20 ± 17.02	484 (97%) 4781 (3%)	0.17 ± 0.03

**Table 3 ijms-23-08756-t003:** IVRT parameters of the indicated formulations, as determined by Franz cells associated with PTFE.

IVRT Parameters	TET_0.9_-DMF	TET_2.7_-DMF	SOL-DMF
R_DMF_ ^1^ ± s.d. (mg/cm^2^/h)	85.28 ± 3.4	63.07 ± 1.9	64.18 ± 1.6
T_lag_ ^2^ ± s.d. (h)	0.41 ± 0.02	0.51 ± 0.10	0.30 ± 0.05
A_DMF_ ^3^ ± s.d. (mg/cm^2^)	175.86 ± 8.2	140.12 ± 10.4	209.39 ± 12.0
M_DMF_ ^4^ ± s.d. (mg)	10.0 ± 2.5	22.5 ± 7.1	15.0 ± 3.2

^1^: DMF release rate; ^2^: lag time; ^3^: amount of DMF released after 8 h; ^4^: M _DMF:_ amount of DMF associated with PTFE membrane after 8 h; DMF was always 500 mg/mL; data are the mean of 6 independent Franz cell experiments ± s.d.

**Table 4 ijms-23-08756-t004:** IVPT parameters of the indicated forms, as determined by Franz cells associated with Strat-M^®^.

IVPT Parameters	TET_0.9_-DMF	TET_2.7_-DMF	SOL-DMF
Jss ^1^ (mg cm^−2^ h^−1^)	16.24 ± 0.7	9.82 ± 0.8	12.05 ± 0.4
T_lag_ ^2^ ± s.d. (h)	0.53 ± 0.01	0.42 ± 0.03	0.55 ± 0.02
Kp ^3^ (cm h^−1^ 10^−3^)	32.48 ± 1.4	19.64 ± 1.6	24.1 ± 0.8
D ^4^ (cm h^−1^) × 10^−3^	0.28 ± 0.1	0.35 ± 0.2	0.27 ± 0.1
P ^5^ _membrane/vehicle_	104.4 ± 5.4	50.5 ± 6.0	80.33 ± 3.0
A_DMF_ ^6^ (mg cm^−2^)	185.2 ± 10.5	112.2 ± 6.8	132.6 ± 9.2
M_DMF_ ^7^ (mg cm^−2^)	25.2 ± 5.2	32.2 ± 6.7	40.4 ± 5.4

^1^: Steady-state flux per unit area, ^2^: lag time; ^3^: permeability coefficient; ^4^: diffusion coefficient; ^5^: partition coefficient; ^6^: cumulative amount of DMF diffused at 24 h; ^7^: DMF associated with the membrane after 24 h; data are the mean of 6 independent Franz cell experiments ± s.d. Differences statistically significant, *p* < 0.05.

## Data Availability

The data presented in this study are available on request from the corresponding author. The data are not publicly available due to privacy restrictions.

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
