# Peer review of "Dimethyl Fumarate-Loaded Transethosomes: A Formulative Study and Preliminary Ex Vivo and In Vivo Evaluation"

_ijms, 2022, doi:10.3390/ijms23158756_

Round 1

Reviewer 1 Report

1-    Include some of the most important numerical results in the abstract.

2-    Why after the introduction, the result is coming in section2? Where is the materials and methods section?

3-    The preparation methodology has no references. Please cite the relevant references.

4-    Why is the release profile not starting from 0% and is showing data from a negative rate? (fig.3)

5-    The Franz cell preparation is not clearly described. What is the membrane and what is the thickness of the membrane?

6-    Why is Tween 80 used in this study? Is not it a toxic material for this issue? Is there any green replacement for this material?

7-    What is the role of xanthan gum?

8-    The introduction can be improved by providing a more critical discussion of recent related literature. Discuss the shortcomings of previous work and the gaps and how this work intends to fill those gaps. For example, some papers  (J. Environ. Treat. Tech7), (211-219, Polymer Testing93, 106922), (Journal of Controlled Release341, 132-146).

Author Response

Response to Reviewer 1 Comments

Point 1-    Include some of the most important numerical results in the abstract.

Response 1- The abstract has been changed accordingly

Point 2-    Why after the introduction, the result is coming in section2? Where is the materials and methods section?

Response 2- Because the Research Manuscript Sections of IJMS follow the order: 1. Introduction, 2. Results, 3. Discussion, 4. Materials and Methods, and 5. Conclusions.

Point 3-    The preparation methodology has no references. Please cite the relevant references.

Response 3- The preparation of transethosomes already referred to reference n° 32 Costanzo, M.; Esposito, E.; Sguizzato, M.; Lacavalla, M.A.; Drechsler, M.; Valacchi, G.; Zancanaro, C.; Malatesta, M. For-mulative study and intracellular fate evaluation of ethosomes and transethosomes for vitamin D3 delivery. Int. J. Mol. Sci. 2021, 22, doi:10.3390/ijms22105341. In the revised manuscript, reference n° 33 has been added: Ferrara, F.; Benedusi, M.; Sguizzato, M.; Cortesi, R.; Baldisserotto, A.; Buzzi, R.; Valacchi, G.; Esposito, E. Ethosomes and Transethosomes as Cutaneous Delivery Systems for Quercetin: A Preliminary Study on Melanoma Cells. Pharmaceutics 2022, 14, doi:10.3390/pharmaceutics14051038.

Point 4-    Why is the release profile not starting from 0% and is showing data from a negative rate? (fig.3)

Response 4- We thank the reviewer for this insightful comment, the slope of the profile had been incorrectly plotted. Accordingly, we revised Figure 3, considering the slope of the release kinetic in the linear part of the profile, starting from 1 h. It should be considered indeed that a lag time characterized all the release profiles. In this respect the rates of the release, as well as T lag, were re-evaluated and reported in Table 3. The graph showing the slopes of the profiles has been added to Supplementary Materials section, as Figure S2.

Point 5-    The Franz cell preparation is not clearly described. What is the membrane and what is the thickness of the membrane?

Response 5- As reported in paragraph 2.6. “In vitro release test (IVRT)”, 2.7. “In vitro permeation test (IVPT)” and 4.6. “Franz cell diffusion experiments”, PTFE membranes were used for IVRT, while STRAT-M® membranes were employed for IVPT. The thickness of the membranes (155.6 mm in the case of PTFE, 316.9 mm in the case of STRAT-M) has been specified in paragraph 4.1. “Materials” (lines 399-340), referring to the new reference https://doi.org/10.1016/j.colsurfb.2019.110613.

Point 6-    Why is Tween 80 used in this study? Is not it a toxic material for this issue? Is there any green replacement for this material?

Response 6- Tween 80 has been previously selected by our group for transethosome production considering different surfactants, evaluating both size distribution and stability (a phrase has been added in the discussion section, lines 349-351). Since reasonably vesicles with smaller size have higher chance to penetrate through the skin over the larger ones, transethosomes based on Tween 80 (0.3%, w/w) were selected, being characterized by higher dimensional stability and smaller mean diameters [26,32,33]. Remarkably, in the present and in previous studies, transethosomes cytoxicity on HaCat cells was investigated by MTT, confirming the safeness of Tween 80 in the formulations. Sodium cholate was investigated in a previous study as green alternative [32], nonetheless the resulting transethosomes were characterized by larger mean diameter, thus we selected Tween 80.

Point 7-    What is the role of xanthan gum?

Response 7- Xanthan gum was employed as a thickening agent, in order to obtain a transethosome gel with a viscosity suitable for cutaneous application, as specified in paragraphs 2.12. “TET gel formulation” (lines 320-321) and 4.10. “Preparation of TET gel” (line 555).

Point 8-    The introduction can be improved by providing a more critical discussion of recent related literature. Discuss the shortcomings of previous work and the gaps and how this work intends to fill those gaps. For example, some papers  (J. Environ. Treat. Tech, 7), (211-219, Polymer Testing, 93, 106922), (Journal of Controlled Release, 341, 132-146).

Response 8- We thank the reviewer for this suggestion, the introduction has been improved accordingly (lines 60-75), considering very recent articles in the field of wound healing technological approaches.

Reviewer 2 Report

The paper entitled "Dimethyl fumarate loaded transethosomes: formulative study and preliminary ex-vivo and in-vivo evaluation" by Francesca Ferrara and co. ,  investigated the  transethosomes  as potential delivery systems for dimethyl fumarate and demonstrated that TET0.9-DMF are characterised by suitable physico-  chemical properties for efficient DMF loading and transdermal penetration.

The subject is interesting and the paper is well written, but some minor revisions must be done, as follows:

1) Pag 3 2.2. Morphology of transethosomes: Please describe briefly why the PC concentration influence the obtaining of unilamellar, multilamellar or  multi vesicular vesicles

2) Pag 3 paragraph 2.3. "Size distribution of transethosomes"  : Please explain brefly why  mean  diameters were influenced by PC and DMF content

Author Response

Response to Reviewer 2 Comments

The paper entitled "Dimethyl fumarate loaded transethosomes: formulative study and preliminary ex-vivo and in-vivo evaluation" by Francesca Ferrara and co. ,  investigated the  transethosomes  as potential delivery systems for dimethyl fumarate and demonstrated that TET0.9-DMF are characterised by suitable physico-  chemical properties for efficient DMF loading and transdermal penetration.

The subject is interesting and the paper is well written, but some minor revisions must be done, as follows:

Point 1- Pag 3 2.2. Morphology of transethosomes: Please describe briefly why the PC concentration influence the obtaining of unilamellar, multilamellar or multi vesicular vesicles

Response 1- It is well known that surfactants and polar lipids spontaneously form different crystalline structures. Particularly, the supramolecular assembly of amphiphiles such as PC, characterized by two  hydrophobic tails and one hydrophilic head group, results in bi-layered vesicles. The shape and size of vesicles is mainly influenced by the molecular packing parameter of the surfactant, a concept based on the geometrical ratio between its tail volume, and the product of the head area and the tail length https://doi.org/10.1021/la010831y. It has been observed that unilamellar vesicles often occur in diluted surfactant solutions, while multivesicular and oligovesicular vesicles are usually found in more concentrated ones. https://doi.org/10.1016/j.cis.2006.01.002. In the case of TET0.9-DMF0.5 the intercalation of the T80 oleate chain within the lipid bilayer could hamper the formation of multilamellar vesicles, while in the case of TET2.7-DMF0.5, a 3-fold increase of PC concentration, under the same amount of T80, can improve the lamellar organization, resulting in multilamellar and multivesicular vesicles. A paragraph has been added at lines 133-137 in section 2.2. and at lines 344-350 in the Discussion section.

Point 2- Pag 3 paragraph 2.3. "Size distribution of transethosomes" : Please explain briefly why  mean  diameters were influenced by PC and DMF content

Response 2- The influence of PC concentration on size of vesicles was previously found by different authors, describing an enlargement of mean diameter due to an increase of PC concentration. doi: 10.1016/j.jconrel.2005.04.007; doi:10.3390/pharmaceutics14051038 . The larger size of TET2.7-DMF0.5 vesicles is due to PC self-organization in multilamellar and oligolamellar vesicles with respect to TET0.9-DMF0.5 unilamellar structure, as observed by cryo-TEM (Figure 1).  The slight increase of vesicle size due to DMF presence could be related to its possible placement within the TET vesicles, near the head groups of PC, at the interface between the glycerol groups and the apolar lipid chains. This positioning can increase the average distance among the PC molecules constituting the bilayer of the vesicles. A paragraph has been added at lines 144-150 in section 2.3. and at lines 352-357 in the Discussion section.